# Plasma and Urinary Metabolomic Analysis of Gout and Asymptomatic Hyperuricemia and Profiling of Potential Biomarkers: A Pilot Study

**DOI:** 10.3390/biomedicines12020300

**Published:** 2024-01-27

**Authors:** Yuki Ohashi, Hiroshi Ooyama, Hideki Makinoshima, Tappei Takada, Hirotaka Matsuo, Kimiyoshi Ichida

**Affiliations:** 1Department of Pathophysiology, Tokyo University of Pharmacy and Life Sciences, Tokyo 192-0392, Japan; y131047@toyaku.ac.jp; 2Department of Pharmacy, International University of Health and Welfare, Tochigi 324-8501, Japan; 3Ryogoku Higashiguchi Clinic, Tokyo 130-0026, Japan; ooyama@higasiguti.jp; 4Tsuruoka Metabolomics Laboratory, National Cancer Center, Yamagata 997-0052, Japan; hmakinos@ncc-tmc.jp; 5Department of Pharmacy, University of Tokyo Hospital, Faculty of Medicine, University of Tokyo, Tokyo 113-8655, Japan; tappei-tky@g.ecc.u-tokyo.ac.jp; 6Department of Integrative Physiology and Bio-Nano Medicine, National Defense Medical College, Saitama 359-8513, Japan; matsuo29@gmail.com; 7Division of Kidney and Hypertension, Department of Internal Medicine, Jikei University School of Medicine, Tokyo 105-8461, Japan; 8Chiba Health Promotion Center, East Japan Railway Company, Chiba 260-0045, Japan

**Keywords:** gout, hyperuricemia, uric acid, ABCG2, SLC2A9, metabolomics

## Abstract

Gout results from monosodium urate deposition caused by hyperuricemia, but most individuals with hyperuricemia remain asymptomatic. The pathogenesis of gout remains uncertain. To identify potential biomarkers distinguishing gout from asymptomatic hyperuricemia, we conducted a genetic analysis of urate transporters and metabolomic analysis as a proof-of-concept study, including 33 patients with gout and 9 individuals with asymptomatic hyperuricemia. The variant allele frequencies of rs72552713, rs2231142, and rs3733591, which are related to serum urate levels (SUA) and gout, did not differ between the gout and asymptomatic hyperuricemia groups. In metabolomic analysis, the levels of citrate cycle intermediates, especially 2-ketoglutarate, were higher in patients with gout than in those with asymptomatic hyperuricemia (fold difference = 1.415, *p* = 0.039). The impact on the TCA cycle was further emphasized in high-risk gout (SUA ≥ 9.0 mg/dL). Of note, urinary nicotinate was the most prominent biomarker differentiating high-risk gout from asymptomatic hyperuricemia (fold difference = 6.515, *p* = 0.020). Although urate transporters play critical roles in SUA elevation and promote hyperuricemia, this study suggests that the progression from asymptomatic hyperuricemia to gout might be closely related to other genetic and/or environmental factors affecting carbohydrate metabolism and urinary urate excretion.

## 1. Introduction

Gout is the most common inflammatory arthritis, and its prevalence is increasing globally [1]. The symptoms of acute inflammatory arthritis (gouty flare) are severe pain, swelling, heat, and joint dysfunction. Extreme pain with gouty flare is rated higher than 7 by patients on a 10-point pain scale, and this pain significantly impairs patients’ quality of life [2]. The pathogenesis of gout involves the formation and deposition of monosodium urate (MSU) crystals, and it is associated with hyperuricemia, which is defined as serum urate levels (SUA) exceeding 7.0 mg/dL, resulting in urate supersaturation in tissues. Therefore, hyperuricemia is a strong predictor of gouty flare. Indeed, the Normative Aging Study reported that the 5-year cumulative incidence of gout increased with increasing SUA (0.5% for SUA ≤ 6.0 mg/dL, 0.6% for SUA = 6.0–6.9 mg/dL, 2.0% for SUA = 7.0–7.9 mg/dL, 4.1% for SUA = 8.0–8.9 mg/dL, 19.8% for SUA = 9.0–9.9 mg/dL, and 30.5% for SUA = ≥10.0 mg/dL) [3]. Nevertheless, 78% of men with baseline SUA ≥ 9.0 mg/dL did not develop gouty flare over the subsequent 5 years [3]. According to a recent large-scale study, including 18,889 participants from four publicly available cohorts, approximately 50% of individuals with SUA ≥ 10.0 mg/dL, well above the physiologic concentration causing urate deposition, developed evident gouty flare over 15 years [4]. In addition, it has been reported that MSU deposition begins in the asymptomatic phase of hyperuricemia [5,6]. These findings suggest that the development of gout cannot be explained by elevated SUA alone, and other additional factors promote the progression of asymptomatic hyperuricemia to gout.

Metabolomics is the field of systems biology that comprehensively investigates metabolites produced by biological activities, and it has rapidly developed in recent years. In various pathological conditions, changes in biological responses attributable to disease cause metabolic changes and alter metabolite profiles specific to the disease. Because changes in metabolic profiles caused by diseases are believed to be reflected in blood and urinary metabolites, metabolomics is considered useful for identifying biomarkers of abnormalities in pathological conditions [7]. Metabolomics has identified pathways and metabolites playing important roles in regulating SUA [8]. To date, some differential metabolites have been identified in studies that compared metabolites between individuals with gout and healthy controls [9,10]. These studies notably identified metabolites and metabolic pathways involved in SUA elevation or gouty flare. Additionally, several metabolomics studies profiled metabolites to predict the progression from asymptomatic hyperuricemia to gout [11,12,13]. However, metabolomic studies in individuals with asymptomatic hyperuricemia and gout only focused on plasma metabolites, and metabolomic analysis of urinary metabolites comparing asymptomatic hyperuricemia to gout has not been conducted. Genome-wide association studies (GWAS) reported a variety of gout-related genes, with renal urate transporters being the most frequently associated genes [14,15]. Thus, urine potentially contains many meaningful metabolites that characterize the onset of gout.

In this study, we investigated the potential for metabolomics to detect differences between gout and asymptomatic hyperuricemia using both plasma and urinary samples as a proof-of-concept (PoC) study.

## 2. Materials and Methods

### 2.1. Study Design and Participants

This study was designed as a cross-sectional observational study. In total, 42 participants (gout, n = 33; asymptomatic hyperuricemia, n = 9) who were outpatients at Ryogoku Higashiguchi Clinic (Tokyo, Japan) were enrolled in this study from September 2019 to April 2020. No patients with COVID-19 were included in this study. All participants were diagnosed with hyperuricemia (SUA > 7.0 mg/dL), and all participants with gout were diagnosed according to the Japanese Guideline on the Management of Hyperuricemia and Gout, 3rd edition [16]. Because most patients with gout are men, the participants in the study, a small-scale PoC trial, were restricted to adult men (age > 20 years).

### 2.2. Data Collection

Data on age, sex, height, weight, body mass index, systolic and diastolic blood pressure, blood parameters (white blood cells, red blood cells, hematocrit, platelets, high-density lipoprotein cholesterol, low-density lipoprotein cholesterol, triglyceride, aspartate aminotransferase, alanine aminotransferase, gamma-glutamyl transpeptidase, hemoglobin A1c, serum creatinine [SCr], SUA, blood urea nitrogen, sodium, potassium, chlorine), and urinary parameters (urinary creatinine [UCr], urinary urate [UUA]) were obtained from patient’s medical records at study entry. We calculated the UUA/UCr ratio and fractional excretion of uric acid (UUA × SCr/SUA × UCr) as the indicators of urinary urate excretion. At the same time, medical questionnaires were used to investigate the durations of hyperuricemia and uric acid-lowering medication use.

### 2.3. Sample Preparation and Metabolomic Measurements Using Gas Chromatography–Tandem Mass Spectrometry (GC-MS/MS)

For GC-MS/MS, 50 μL of plasma and urine samples were collected from participants with gout or asymptomatic hyperuricemia. The samples, which were mixed with 10 μL (0.5 mg/mL) of 2-isopropylmalic acid (Sigma-Aldrich, St. Louis, MO, USA) as an internal standard, were additionally mixed with 1 L of a solvent mixture (MeOH/H_2_O/CHCl_3_ = 2.5/1/1, *v*/*v*/*v*). The mixture was centrifuged at 1200 rpm for 30 min at 37 °C followed by 16,000× *g* for 3 min at 4 °C. Then, a 600-μL aliquot of the supernatant was transferred to a clean tube, and 300 μL of water was added to the tube. After mixing, the solution was centrifuged at 16,000× *g* for 3 min at 4 °C. Next, 400 μL of the supernatant was transferred to a clean tube and then dried overnight using a spin dryer (TAITEC, Koshigaya, Japan). After drying, 100 μL (20 mg/mL) of methoxyamine hydrochloride in pyridine was added, and the resulting mixture was incubated for 90 min at 30 °C. For derivatization, 40 μL of MSTFA was added, and the mixture was incubated for 30 min at 37 °C.

The derivatized samples were eluted with 100 μL of n-hexane, and 1 μL of the derivatized solution was injected into a GC-MS device (GCMS-TQ8050, Shimadzu, Kyoto, Japan). Helium was used as a carrier gas at a constant flow rate of 39.0 cm/s. The transfer line and ion source temperatures were 280 and 200 °C, respectively. For MS, electron ionization was performed at 70 eV, and argon was used as a collision-induced dissociation gas. Metabolite detection was performed using the Smart Metabolites Database and GCMS Solution software (Ver. 4.45, Shimadzu, Kyoto, Japan).

### 2.4. Genetic Analysis

Approximately 5 mL of blood collected from each participant was centrifuged to isolate leukocytes (3500 rpm, 20 °C, 30 min). Thereafter, a lysis buffer containing 3 μg/mL proteinase K was added to the cells. After extraction with TE-saturated phenol, ethanol precipitation was performed to purify the genomic DNA. 

In total, 2 exons of ATP-binding cassette subfamily G member 2 (*ABCG2*) and 13 exons of solute carrier family 2 member 9 (*SLC2A9*/*GLUT9* variants 1 and 2) were analyzed. To analyze two common SNPs of *ABCG2* (rs72552713 and rs2231142), we performed TaqMan-based probe qualitative real-time PCR for genotyping using a THUNDERBIRD Probe qPCR Mix kit (Toyobo, Osaka, Japan) and PikoReal 96 system (Thermo Fisher Scientific, Waltham, MA, USA) using 40 ng of extracted genomic DNA according to the manufacturer’s protocols. To sequence *SLC2A9* exons, 100 ng of genomic DNA were amplified in a 50-μL volume containing 1.25 U of PrimeSTAR GXL DNA Polymerase (TaKaRa Bio, Siga, Japan), 200 μM dNTPs, and the primer pair (0.4 μM). Amplified PCR products were purified using 1% agarose gel in 1 × TAE buffer and a QIAquick Gel Extraction Kit (QIAGEN, Hilden, Germany). DNA sequencing was performed using an Applied Biosystems 3730xl DNA Analyzer (Thermo Fisher Scientific, Waltham, MA, USA).

### 2.5. Statistical Analysis

All participants (N = 42) were classified into three groups: high-risk gout (SUA ≥ 9.0 mg/dL. n = 16), low-risk gout (SUA < 9.0 mg/dL, n = 17), and asymptomatic hyperuricemia (n = 9). The cutoff for SUA was set at 9.0 mg/dL [3], as the risk of gout is believed to dramatically increase at SUA levels of 9.0 mg/dL or higher. Each continuous value was presented as the median unless otherwise noted. Participant characteristics (demographic, genetic, clinical, and metabolic characteristics) were compared using the Kruskal–Wallis test, Steel’s test, or Fisher’s exact test.

First, multivariate partial least squares (PLS) regression analysis was performed to select candidate metabolites associated with the progression from asymptomatic hyperuricemia to gout. After separating participants with gout and asymptomatic hyperuricemia by PLS regression, the metabolites that contributed to the separation were selected. The predictors (X) and response (Y) were metabolites detected by GC-MS/MS and participant classification (high-risk gout, low-risk gout, and asymptomatic hyperuricemia), respectively. The PLS components were constrained to be orthogonal. The dimensionality-reducing transformation builds a matrix in which columns represent the first P eigenvectors of the matrix formed by the covariance between X and Y [17]. Hence, PLS regression selected a subset of scores and loadings that most effectively summarized X and Y and described the correlation between them [17]. However, PLS-based methods cannot provide any statistical significance of variables expressed by *p*-values. Instead, we calculated the variable importance for projection (VIP), which reflects the influence of each variable on the response. In this study, we identified 30 metabolites with the highest VIP scores as candidate metabolites associated with the pathogenesis of gout derived from PLS regression.

Second, the univariate Mann–Whitney U test and Steel’s test were conducted for candidate metabolites, and fold differences of medians compared to the asymptomatic hyperuricemia group were calculated. *p*-values were adjusted using the Benjamini–Hochberg method to control the false discovery rate (FDR) [18]. To include any potential metabolites as comprehensively as possible, we employed adjusted *p* < 0.15 as the threshold for potential biomarkers distinguishing gout from asymptomatic hyperuricemia. For multivariate and univariate analyses, JMP Pro 15 (SAS Institute Inc., Cary, NC, USA) was used. There were no missing clinical parameters, and no detected metabolites by GC-MS/MS were calculated as zero.

Then, pathway analysis of the potential biomarkers was applied within MetaboAnalyst 5.0 (https://www.metaboanalyst.ca (accessed on 13 January 2023)). Pathway analysis was performed using the KEGG (http://www.kegg.jp (accessed on 13 January 2023)) *Homo sapiens* pathway library and the hypergeometric test. Among all perturbed pathways, those with an impact value > 0.1 and adjusted *p* < 0.05 were selected as perturbed metabolic pathways associated with the development of gout. The metabolites were mapped into the pathways associated with each significant metabolite.

## 3. Results

### 3.1. Demographic, Clinical, and Genetic Characteristics of the Enrolled Participants

The demographic characteristics (e.g., uric acid-lowering medication use, complications, and familial history of gout) of the participants, which were obtained from the questionnaires, are presented in Table 1. Six participants with gout were on uric acid-lowering therapies, namely xanthine oxidase (also known as xanthine oxidoreductase) inhibitors (allopurinol, febuxostat, or topiroxostat). Several participants with gout had urolithiasis and hypertension, whereas no participants with asymptomatic hyperuricemia had coincident diseases. A familial history of gout was more common in the asymptomatic hyperuricemia group than in the gout group (55.6% vs. 24.2%).

The clinical characteristics of the participants are presented in Table 2. All participants had SUA > 8.0 mg/dL (range, 8.1–10.7 mg/dL). Participants with gout were older than those with asymptomatic hyperuricemia. The duration of hyperuricemia was 4.0 (3.5–9.0), 7.0 (5.0–10.0), and 9.0 (4.1–13.8) years in the asymptomatic hyperuricemia, low-risk gout, and high-risk gout groups, respectively. Urinary uric acid excretion and fractional excretion of uric acid were low among the study participants, and they tended to be lower in participants with gout, albeit without significance.

The common dysfunctional variants of *ABCG2* (rs72552713 and rs2231142) and synonymous (rs13113918, rs10939650, rs3733589, rs13125646) and non-synonymous variants (rs6820230, rs2276961, rs3733591, rs2280205) of *SLC2A9* were analyzed (Table 3). The frequencies of *ABCG2* dysfunctional variants were high among the participants, and they tended to be higher in participants with asymptomatic hyperuricemia than in those with gout. 

### 3.2. Multivariate PLS Discriminant Analysis (PLS-DA) to Select the Candidate Metabolites That Distinguish Gout from Asymptomatic Hyperuricemia

GC-MS/MS identified 149 metabolites in plasma and 211 metabolites in urine, and 135 metabolites were commonly detected in both plasma and urine samples (Appendix A). To extract the candidate metabolites associated with the development of gout, we performed PLS-DA to distinguish participants with gout from those with asymptomatic hyperuricemia.

First, we conducted PLS-DA between the high-risk gout and asymptomatic hyperuricemia groups. As presented in Figure 1a, high-risk gout samples could be distinguished from asymptomatic hyperuricemia samples. As illustrated in Figure 1b, 30 metabolites contributing to the differentiation of the groups were extracted as the candidate metabolites. Among these 30 metabolites, the metabolites that notably contributed to the discrimination (VIP > 2.0) were nonanoate, eicosapentaenoate, 2-ketoglutarate, arachidonate, and urate in plasma and nicotinate and sucrose in urine. Then, we conducted PLS-DA between low-risk gout and asymptomatic hyperuricemia groups (Figure 1c), and 30 candidate metabolites were extracted (Figure 1d). The metabolites that notably contributed to the discrimination (VIP > 2.0) were glucose in plasma and nicotinate, sucrose, and paraxanthine in urine. Of the 30 candidate metabolites extracted in both analyses, 10 common metabolites were identified: 1,5-anhydroglucitol, 2-ketoglutarate, glucose, eicosapentaenoate, nonanoate, and phosphorate in plasma and nicotinate, sucrose, creatinine, and hydroxylamine in urine. Finally, 49 metabolites were selected for the subsequent univariate and metabolic pathway analyses.

### 3.3. Univariate and Metabolic Pathway Analyses of Gout and Asymptomatic Hyperuricemia

To further extract the potential biomarkers distinguishing gout from asymptomatic hyperuricemia, we initially summarized the 49 candidate metabolites and tested each candidate metabolite for increasing or decreasing trends between gout and asymptomatic hyperuricemia groups (Table 4). In particular, there were clear increasing trends in plasma 2-ketoglutarate (raw *p* < 0.001) and urinary nicotinate levels (raw *p* = 0.001) in the gout group compared to the findings in the asymptomatic hyperuricemia group.

We used the univariate Mann–Whitney U test to analyze 49 candidate metabolites identified via PLS-DA. Twenty-six potential biomarkers satisfying FDR = 0.15 were found in univariate analysis (Table 4). In the gout group, 2-ketoglutarate (fold difference = 1.415, raw *p* < 0.001) and nicotinate (fold difference = 5.475, raw *p* = 0.001) were especially accumulated and excessively excreted compared to the findings in the asymptomatic hyperuricemia group (Table 5). 

Furthermore, Steel’s test was conducted to compare the levels of the 49 candidate metabolites among the asymptomatic hyperuricemia, low-risk gout, and high-risk gout groups. Eighteen potential biomarkers discriminating high-risk gout and asymptomatic hyperuricemia that satisfied the FDR threshold were found, but no potential biomarker discriminating low-risk gout and asymptomatic hyperuricemia was identified (Table 5). 

According to the results of pathway analysis using potential biomarkers distinguishing gout from asymptomatic hyperuricemia, the TCA cycle (impact = 0.200, adjusted *p* = 0.003) was perturbed (Figure 2a and Appendix A). Additionally, we performed pathway analysis using potential biomarkers to distinguish high-risk gout from asymptomatic hyperuricemia (Figure 2b and Appendix A). In the high-risk gout group, the impact on the TCA cycle in gout was further emphasized (impact = 0.512, adjusted *p* < 0.001). In addition to the TCA cycle, glyoxylate and dicarboxylate metabolism (impact = 0.182, adjusted *p* = 0.006); alanine, aspartate, and glutamate metabolism (impact = 0.333, adjusted *p* = 0.006); and pyruvate metabolism (impact = 0.455, adjusted *p* = 0.031) were identified as gout-related metabolic pathways (Figure 2b and Appendix A). We mapped the perturbed pathway and compared potential biomarkers among the high-risk gout, low-risk gout, and asymptomatic hyperuricemia groups (Figure 2c,d).

## 4. Discussion

Gout progression can be defined in gradual pathological stages: asymptomatic hyperuricemia without evidence of MSU crystal deposition, asymptomatic hyperuricemia with evidence of MSU deposition, MSU deposition with prior or current symptoms of acute gouty flares, and advanced gout characterized by tophi and chronic gouty arthritis [19]. The pathophysiology of gout is based on the deposition of MSU crystals, which form in the presence of elevated SUA; i.e., hyperuricemia is the most critical predictor of gout. However, gouty flares do not develop in all individuals with hyperuricemia, and most cases of hyperuricemia remain asymptomatic [3,4]. Epidemiological evidence indicates that there are modifying factors in the stages from normouricemia to asymptomatic hyperuricemia and from asymptomatic hyperuricemia to gout. In this study, a GC-MS/MS-based metabolomic investigation using both plasma and urine samples was conducted as a PoC study to identify differential features between gout and asymptomatic hyperuricemia.

Gout is associated with both environmental and genetic factors [20,21,22,23,24]. GWAS and epidemiological studies identified several genes associated with gout [14,15,20], including the two most prominent urate transporters (ABCG2 and SLC2A9). Most GWAS compared gout to normouricemia (or without strict separation from asymptomatic hyperuricemia), and few GWAS investigated genetic factors associated with the progression from asymptomatic hyperuricemia to gout [25]. Previous GWAS using participants with gout and asymptomatic hyperuricemia detected the loci of genes encoding urate transporters (*ABCG2* and *SLC2A9*) as genetic factors aggravating normouricemia into gout [25]. In addition, three genes (*CNTN5*, *MIR302F*, and *ZNF724*) were detected as ‘gout vs. asymptomatic hyperuricemia’-specific genetic factors [25]. This suggests that mechanisms other than SUA elevation are involved in the progression from asymptomatic hyperuricemia to gout. In this study, we also investigated *ABCG2* and *SLC2A9* in participants with gout and asymptomatic hyperuricemia. Although the variant allele frequencies of the dysfunctional SNPs of *ABCG2* (rs72552713 and rs2231142) were high, there were no differences in the frequencies of *ABCG2* variants associated with SUA elevation between the gout and asymptomatic hyperuricemia groups (Table 3). The pathogenic variants of *SLC2A9* were assessed in all participants, and the frequency of rs3733591, which has been reported to be related to SUA [26,27,28], was not significantly different between the gout and asymptomatic hyperuricemia groups. This result suggests that other factors, including other urate transporters and environmental factors, contribute to the progression from asymptomatic hyperuricemia to gout, although the urate transporters ABCG2 [20,29,30,31] and SLC2A9 [32] play an important role in regulating SUA.

Additionally, a previous network analysis identified a cluster of genes involved in glucose metabolism as gout-related factors [14]. Our metabolomic analysis also suggested that carbohydrate metabolism is enhanced in high-risk gout (Figure 2a,b). This result was consistent with that of the previous study. Several studies found that plasma glucose is positively and negatively related to SUA and urinary urate clearance, respectively, via insulin resistance [33,34,35,36]. Insulin is the hormone that promotes the utilization of glucose and lowers blood glucose levels. Although insulin concentrations were not measured in this study, the levels of glucose, TCA cycle intermediates, and pyruvate and lactate, which are the main products of glycolysis, tended to be higher in patients with gout. In particular, lactate is interpreted as a marker of typical anaerobic glycolysis, and its accumulation usually implies a high energy demand in biological systems; e.g., glycolysis activation accompanied by hypoxia or inflammation increases the accumulation of lactate [37]. In the present study, plasma lactate levels tended to be higher in the gout group, suggesting that the accumulation of lactate reflects the low utilization and/or increased intake of glucose and facilitates the induction of gouty flare via decreased pH. In the gout group, 2-ketoglutarate, isocitrate, and malate tended to accumulate compared to the findings in the asymptomatic hyperuricemia group (Table 4 and Table 5). The accumulation of these TCA cycle intermediates requires the reduction of nicotinamide adenine dinucleotide (NAD+) for subsequent reductions. By contrast, the plasma levels of other TCA cycle intermediates (citrate, succinate, and fumarate) that do not require NAD+ reduction were not significantly different between gout and asymptomatic hyperuricemia groups (Appendix A). This imbalance also suggests that gout onset is at least partially related to carbohydrate metabolism, including mitochondrial internal respiration. Acute gouty flare is initiated by synovial resident macrophages, which are innate immune cells. The deposition of MSU crystals induces nucleotide-binding domain, leucine-rich-containing family, and pyrin domain-containing 3 (NLRP3) inflammasome formation and mediates localized acute inflammation [38]. In addition to localized inflammation, NLRP3 inflammasome activation leads to a persistent systemic inflammatory response, including cardiovascular dysfunction. In epidemiological studies, high SUA levels were linked to cardiovascular dysfunction [39,40,41]. Indeed, gout flares in the preceding days have been reported to increase the risk of cardiovascular events [42]. Although the association between mitochondria function and NLRP3-associated inflammations, including gout and cardiovascular events, has been reported [43,44,45,46], it remains controversial because NAD+, which reflects mitochondrial internal respirational function, was not measured in this study.

It has been reported that the main pathology of hyperuricemia in patients with gout is decreased urate excretion in the kidneys and intestine [29,47]. The regulation of urate excretion, especially by a cluster of urate transporters in the kidneys, is important in the homeostasis of SUA [48], which has been corroborated by the presence of type I and type II renal hypouricemia caused by SLC22A12 [49,50,51] and SLC2A9 dysfunction [32,52], respectively. In urate reabsorption, SLC22A12, which is expressed at the apical membrane of proximal tubule cells, transports urate inside the cell in conjunction with the outward transport of monocarboxylate (Figure 2d). In addition, SLC5A8 transports monocarboxylate, which is essential as the counterpart in urate reabsorption, from urine into proximal tubule cells [53]. Namely, SLC22A12 and SLC5A8 act cooperatively in urate reabsorption mediated by monocarboxylate. Indeed, it has been reported that excess blood lactate is excreted in exchange for reabsorbed urate [54]. Of note, urinary nicotinate was identified as the metabolite that most prominently differentiated gout from asymptomatic hyperuricemia in this study (Table 4 and Table 5). Although we cannot discuss causality because this was a non-interventional and cross-sectional observation, this result might reflect the association between gout and the accumulation of monocarboxylates, including lactate and nicotinate, the renal excretion of which is coupled to urate reabsorption.

However, some limitations of the present study must be acknowledged, and several problems and issues must be addressed for future full-scale metabolomics studies. First, the results of this study were based on a small sample size, limiting the robustness of the findings. In particular, the sample size must be increased to facilitate the identification of potential biomarkers distinguishing low-risk gout from asymptomatic hyperuricemia. The reason is that the pathophysiology of gout is based on asymptomatic hyperuricemia, and the metabolic features of these conditions are similar. In addition, our results were not adjusted for other effect modifiers, including age. Future full-scale metabolomics studies should be conducted with sufficient sample sizes permitting adjustment for effect modifiers, including age. The second is consideration of the duration of hyperuricemia. Naturally, the cumulative risks of progression to gout increase with the duration of hyperuricemia [3]. Therefore, a more sensitive discriminant analysis might be feasible by comparing individuals who remain asymptomatic despite a long duration of hyperuricemia and patients with gout who experience flares despite a short duration of hyperuricemia. Because this study did not include many such individuals, this is proposed as a future issue. Third, genetic factors other than urate transporters should be investigated. Although we found that carbohydrate metabolism was perturbed in patients with high-risk gout, we did not clarify whether this is attributable to genetic or environmental factors. Our pilot study identified glucose metabolism factors that have been implicated in gout as genetic factors to be investigated in future full-scale studies. Finally, we must acknowledge the limitations of our results regarding confounders and generalization. Six patients with gout received xanthine oxidase inhibitors within 2 weeks of enrollment. Xanthine oxidase inhibitors have been reported to reduce the incidence of cardiovascular events as well as gout flares [55,56,57]. It is undeniable that this external intervention might have confounded the results (e.g., inflammatory biomarker levels) of this study. In addition, the entire discovery dataset in this study consisted of data from only men, which limits the generalizability of the findings. Despite these limitations, this study comprehensively profiled both plasma and urinary metabolism and attempted to differentiate gout and asymptomatic hyperuricemia. In conclusion, our pilot study suggests that glycolysis compounds, TCA cycle intermediates, and urinary nicotinate, which are related to urinary urate excretion, are potential biomarkers distinguishing gout from asymptomatic hyperuricemia. 

## Figures and Tables

**Figure 1 biomedicines-12-00300-f001:**
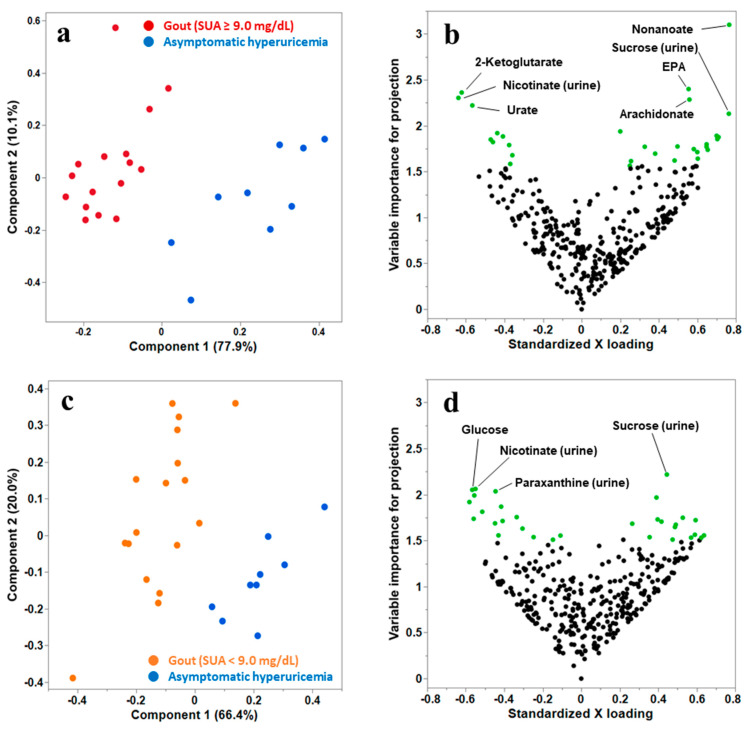
(**a**) The PLS scatter plot differentiates participants with high-risk gout [serum urate levels (SUA) ≥ 9.0 mg/dL] from those with asymptomatic hyperuricemia. Component 1 can explain 77.9% of the variance (gout or asymptomatic hyperuricemia), and component 2 can explain 10.1% of the variance. (**b**) The volcano plot of plasma and urinary samples from participants with high-risk gout (SUA ≥ 9.0 mg/dL) and asymptomatic hyperuricemia. The vertical and horizontal axes present the variable importance for projection (VIP) and standardized loading value of the metabolites for component 1, respectively. The 30 metabolites with the highest VIP values are plotted in green. (**c**) The PLS scatter plot differentiates participants with low-risk gout (SUA < 9.0 mg/dL) from those with asymptomatic hyperuricemia. Component 1 can explain 66.4% of the variance, and component 2 can explain 20.0% of the variance. (**d**) The volcano plot of plasma and urinary samples from participants with low-risk gout (SUA < 9.0 mg/dL) and asymptomatic hyperuricemia. Vertical and horizontal axes denote the VIP and standardized loading value of the metabolites for component 1. The 30 metabolites with the highest VIP values are plotted in green.

**Figure 2 biomedicines-12-00300-f002:**
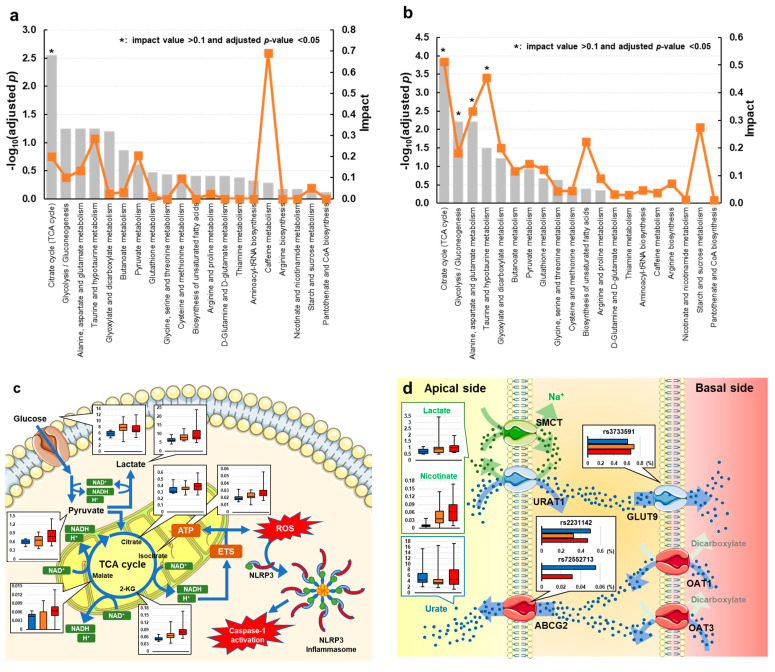
Summary of the effects of metabolic pathways using potential biomarkers distinguishing (**a**) overall gout and (**b**) gout with SUA ≥ 9.0 mg/dL from asymptomatic hyperuricemia. The *x*-axis displays the pathways, and the *y*-axis displays the −log of the adjusted *p*-value calculated by the hypergeometric test and adjusted by the Benjamini–Hochberg method to control the false discovery rate. Orange dots connected by a thin line represent the impact. Perturbed metabolism in (**c**) the intracellular TCA cycle and (**d**) the mechanism of uric acid excretion in the renal tubules. Red, orange, and blue box plots and bar plots denote high-risk gout (SUA ≥ 9.0 mg/dL), low-risk gout (SUA < 9.0 mg/dL), and asymptomatic hyperuricemia groups, respectively. Box plots show the lower and upper quartiles and the median, representing the maximum to minimum range (AU). Bar plots present the variant allele frequencies (%) of rs2231142, rs72552713, and rs3733591. Parts of the figure were drawn by using pictures from Servier Medical Art. Servier Medical Art by Servier is licensed under a Creative Commons Attribution 3.0 Unported License (https://creativecommons.org/licenses/by/3.0/ (accessed on 13 January 2023)). Abbreviation: SUA, serum urate levels.

**Table 1 biomedicines-12-00300-t001:** Primary demographic characteristics of participants with gout and asymptomatic hyperuricemia.

					Gout (n = 33)						*p*
	All (n = 42)		AHU (n = 9)		All Gout		Low-Risk Gout (SUA < 9.0 mg/dL)		High-Risk Gout (SUA ≥ 9.0 mg/dL)		
	n	%	n	%	n	%	n	%	n	%	
** Demographic data **											
Family history of gout ^1^	13	31.0	5	55.6	8	24.2	2	11.8	6	37.5	0.056
Uric acid-lowering treatment ^2^	6	14.3	0	0	6	18.2	2	11.8	4	25	0.232
Urolithiasis	6	14.3	0	0	6	18.2	4	23.5	2	12.5	0.339
Hypertension	6	14.3	0	0	6	18.2	4	23.5	2	12.5	0.339
Hyperlipidemia	4	38	1	11.1	3	9.1	2	11.8	1	6.3	1

Abbreviations: AHU, asymptomatic hyperuricemia; SUA, serum urate levels. ^1^ Family history of gout is limited to the second degree of consanguinity. ^2^ Doses (xanthine oxidase inhibitors) taken within 2 weeks were counted. Fisher’s exact test was used to compare the AHU and gout groups.

**Table 2 biomedicines-12-00300-t002:** Clinical and biochemical characteristics of participants with gout and asymptomatic hyperuricemia.

			Gout			Kruskal–Wallis
Median (IQR: Q1–Q3)	All	AHU	All Gout	Low-Risk Gout (SUA < 9.0 mg/dL)	High-Risk Gout (SUA ≥ 9.0 mg/dL)	*p*
**Basic characteristics**						
Age (years)	45 (39–55)	34 (28–44)	50 (40–57)	50 (43–58) ^a^	47 (39–57) ^a^	**0.005**
Duration of hyperuricemia (years)	7.0 (4.0–10.0)	4.0 (3.5–9.0)	8.0 (4.7–10.0)	7.0 (5.0–10)	9.0 (4.1–13.8)	0.357
Body mass index (kg/m^2^)	25.6 (23.6–28.3)	26.2 (23.6–29.6)	25.6 (23.6–27.6)	24.6 (23.2–26.1)	26.9 (24.0–30.1)	0.117
**Clinical characteristics**						
SBP (mmHg)	124 (120–134)	122 (119–134)	124 (120–130)	122 (120–129)	127 (119–140)	0.487
DBP (mmHg)	82 (76–87)	78 (71–85)	82 (77–88)	82 (77–86)	82 (77–94)	0.697
WBCs (10^3^/μL)	66 (57–79)	61 (51–69)	68 (57–81)	63 (55–77)	71 (58–88)	0.229
RBCs (10⁶/μL)	483 (463–520)	522 (517–533)	477 (449–490)	479 (454–496) ^a^	475 (444–523) ^a^	**<0.001**
Hematocrit (%)	45.2 (42.8–47.0)	46.1 (45.7–48.4)	45 (42.0–46.5)	45.0 (42.6–46.7)	43.1 (41.5–46.5) ^a^	**0.034**
PLTs (10^3^/μL)	24.7 (22.5–29.8)	23.7 (20.8–25.0)	27.6 (23.2–31.1)	23.8 (22.2–30.1)	28.8 (23.3–32.7)	0.107
AST (IU/L)	25 (20–32)	30 (20–37)	25 (20–32)	27 (24–25)	21 (19–28)	0.115
ALT (IU/L)	29 (21–42)	40 (21–68)	28 (21–39)	25 (22–38)	29 (21–44)	0.371
γ-GTP (IU/L)	53 (29–81)	77 (35–186)	49 (28–71)	57 (27–68)	45 (37–76)	0.334
LDL-C (mg/dL)	130 (104–159)	122 (95–190)	130 (108–152)	132 (106–145)	128 (108–167)	0.892
HDL-C (mg/dL)	53 (46–65)	49 (42–55)	53 (46–67)	57 (46–72)	53 (46–65)	0.184
TG (mg/dL)	147 (88–201)	178 (131–257)	138 (84–183)	102 (81–181)	151 (122–200)	0.149
BUN (mg/dL)	12.5 (10.1–14.9)	13.1 (12.4–14.8)	12.3 (9.8–15.1)	12.1 (9.6–14.3)	12.9 (11.3–15.8)	0.271
HbA1c	5.4 (5.2–5.6)	5.3 (4.9–5.6)	5.4 (5.2–5.7)	5.4 (5.2–5.6)	5.5 (5.2–5.8)	0.207
Sodium (mEq/L)	140 (138–141)	140 (138–142)	140 (138–141)	139 (138–140)	141 (138–142)	0.239
Potassium (mEq/L)	4.2 (4.0–4.4)	4.3 (4.1–4.4)	4.2 (4.0–4.5)	4.2 (4.0–4.4)	4.4 (4.0–4.6)	0.495
Chloride (mEq/L)	102 (101–104)	101 (100–103)	102 (102–105)	102 (100–105)	103 (102–104)	0.144
eGFR (mL/min/1.73 m^2^)	76.5 (63.8–83.7)	78.9 (70.8–97.3)	72.8 (63.0–81.0)	76.4 (64.7–84.1)	71.6 (60.2–78.7)	0.143
SUA (mg/dL)	8.8 (8.5–9.5)	8.8 (8.6–9.4)	8.8 (8.4–9.7)	8.4 (8.2–8.6) ^a^	9.7 (9.4–10.5) ^a,b^	**<0.001**
UUA (mg/dL)	9 (7–13)	9 (8–31)	9 (7–13)	8 (6–12)	10 (8–14)	0.367
UUA/UCr ratio	0.42 (0.33–0.50)	0.46 (0.39–0.50)	0.39 (0.31–0.50)	0.41 (0.32–0.50)	0.38 (0.31–0.50)	0.51
FEUA (%)	4.1 (3.3–4.8)	4.4 (3.9–4.8)	4.0 (3.2–5.1)	4.2 (3.3–5.4)	3.6 (2.9–4.8)	0.33

Abbreviations: AHU, asymptomatic hyperuricemia; BMI, body mass index; SBP, systolic blood pressure; DBP, diastolic blood pressure; WBCs, white blood cells; RBCs, red blood cells; PLTs, platelets; AST, aspartate aminotransferase; ALT, alanine aminotransferase; γ-GTP, γ-glutamyl transferase; TG, triglyceride (mg/dL to mmol/L, ×0.01129); HDL-C, high-density lipoprotein cholesterol (mg/dL to mmol/L, ×0.02586); LDL-C, low-density lipoprotein cholesterol (mg/dL to mmol/L, ×0.02586); BUN, blood urea nitrogen (mg/dL to mmol/L, ×0.357); SUA, serum urate levels (mg/dL to μmol/L, ×59.48); UUA, urinary urate (mg/dL to μmol/L, ×59.48); UCr, urinary creatinine (mg/dL to μmol/L, ×88.4); eGFR, estimated glomerular filtration (mL/min/1.73 m^2^; 194 × serum creatinine^−1.094^ × age^−0.287^ [× 0.739 if female]); FE_UA_, fractional excretion of uric acid (%; urate clearance/creatinine clearance). Differences between participants with AHU and participants with gout were compared using the Kruskal–Wallis test. Statistical significance was determined using the Steel test: ^a^
*p* < 0.05 versus the AHU group, ^b^
*p* < 0.05 versus the low-risk gout group.

**Table 3 biomedicines-12-00300-t003:** Genetic characteristics of participants with gout and asymptomatic hyperuricemia.

			Gout			
		AHU	Overall Gout	Low-Risk Gout (SUA < 9.0 mg/dL)	High-Risk Gout (SUA ≥ 9.0 mg/dL)	
Variant	rsID	VAF	VAF	VAF	VAF	Ref. VAF
***ABCG2* SNPs**						
** Non-synonymous variants**						
p.Q126X, c.376C>T	rs72552713	0.056	0.015	0	0.031	0.009
p.Q141K, c.421C>A	rs2231142	0.500	0.394	0.324	0.469	0.291
***SLC2A9* SNPs**						
** Non-synonymous variants**						
p.A17T, c.49G>A	*rs6820230*	0.222	0.091	0.118	0.063	0.074
p.G25R, c.73G>A	*rs2276961*	0.333	0.424	0.412	0.438	0.451
p.R294H, c.881G>A	*rs3733591*	0.611	0.682	0.706	0.656	0.686
p.P350L, c.1049C>T	*rs2280205*	0.111	0.288	0.176	0.406	0.286
** Synonymous variants**						
p.L108L, c.322T>C	*rs13113918*	1	0.970	0.941	1	0.975
p.T125T, c375G>A	*rs10939650*	0.611	0.515	0.559	0.469	0.488
p.I168I, c.504C>T	*rs3733589*	0.333	0.470	0.471	0.469	0.401
p.L189L, c.567T>C	*rs13125646*	1	1	1	1	0.977

Abbreviations: SUA, serum urate levels (mg/dL to μmol/L, ×59.48); VAF, variant allele frequency; *ABCG2*, ATP-binding cassette subfamily G member 2; *SLC2A9*, solute carrier family 2 member 9. Ref. VAF refers to the East Asian VAF in Allele Frequency Aggregator (ver. 20201027095038).

**Table 4 biomedicines-12-00300-t004:** Potential biomarkers distinguishing asymptomatic hyperuricemia from overall gout.

		Median (IQR: Q1–Q3)					
Metabolites (Arbitrary Units)	Sample	All (n = 42)	AHU (n = 9)	Gout (n = 33)	Trend	Raw *p*	Adjusted *p*
**Glucose metabolites**							
Glucose	Plasma	7.02 (5.98–8.67)	5.88 (4.93–6.75)	7.43 (6.35–9.07)	↑	0.006	0.074
Pyruvate	Plasma	0.67 (0.57–0.81)	0.60 (0.54–0.67)	0.70 (0.60–0.91)	↑	0.048	0.124
Lactate	Plasma	7.42 (6.38–9.41)	6.28 (5.63–7.71)	7.86 (6.77–10.00)	↑	0.018	0.080
1,5-Anhydroglucitol (×10^−2^)	Plasma	0.63 (0–1.00)	0.86 (0.67–1.48)	0.54 (0–0.95)	↓	0.037	0.121
2-Phosphoglycerate (×10^−3^)	Urinary	0 (0–0)	0 (0–6.60)	0 (0–0)	-	0.125	0.180
**TCA cycle intermediates**							
Aconitate (×10^−3^)	Plasma	0 (0–1.70)	0 (0–0)	0 (0–7.46)	-	0.065	0.127
Isocitrate (×10^−2^)	Plasma	2.18 (1.96–2.73)	1.89 (1.62–2.13)	2.34 (2.07–2.86)	↑	0.015	0.092
2-Ketoglutarate (×10^−2^)	Plasma	7.04 (5.53–8.62)	5.18 (4.61–5.85)	7.33 (6.45–9.01)	↑	<0.001	0.039
Malate (×10^−3^)	Plasma	5.02 (0–6.70)	4.62 (0–5.33)	5.51 (0–7.05)	-	0.349	0.372
**Lipid metabolites**							
Arachidonate (×10^−1^)	Plasma	0.72 (0.52–0.86)	0.85 (0.73–1.02)	0.67 (0.43–0.83)	↓	0.015	0.082
Eicosapentaenoate (×10^−2^)	Plasma	7.33 (5.33–8.83)	9.31 (7.38–9.53)	7.18 (4.92–8.55)	↓	0.013	0.091
Docosahexaenoate (×10^−2^)	Plasma	1.20 (0–1.78)	0.55 (0–1.45)	1.25 (0.38–1.93)	-	0.117	0.179
Caproate (×10^−1^)	Plasma	1.18 (1.03–1.39)	1.30 (1.07–1.41)	1.17 (0.95–1.41)	-	0.350	0.365
Nonanoate (×10^−2^)	Plasma	3.63 (2.98–6.88)	6.84 (6.27–7.29)	3.31 (2.67–5.93)	↓	0.004	0.065
**Amino acid metabolites**							
Tryptophan	Plasma	1.22 (0.93–1.74)	1.58 (1.19–2.20)	1.11 (0.85–1.70)	-	0.055	0.128
5-Oxoproline	Plasma	7.17 (5.85–7.84)	7.77 (6.71–8.75)	6.86 (5.72–7.71)	-	0.064	0.131
Serine	Plasma	0.48 (0.19–1.02)	0.98 (0.46–1.12)	0.42 (0.18–0.87)	-	0.083	0.151
Cysteine (×10^−2^)	Plasma	3.12 (2.46–4.51)	2.58 (1.62–3.01)	3.22 (2.59–5.06)	-	0.055	0.123
Lysine	Plasma	0.57 (0.36–0.86)	0.59 (0.51–0.97)	0.55 (0.35–0.81)	-	0.319	0.364
Tyramine	Plasma	1.20 (0.71–1.67)	1.27 (0.97–1.91)	1.15 (0.68–1.63)	-	0.366	0.374
Aspartate (×10^−2^)	Urinary	4.35 (2.59–7.91)	6.32 (4.82–9.39)	3.85 (2.47–7.50)	-	0.122	0.181
Tyramine (×10^−1^)	Urinary	2.36 (1.69–3.80)	3.74 (1.94–7.25)	2.34 (1.63–3.61)	-	0.145	0.197
4-Aminobutyrate (×10^−2^)	Urinary	4.24 (0–7.43)	7.35 (3.82–9.97)	3.63 (0–6.28)	-	0.051	0.125
5-Aminolevulinate (×10^−2^)	Urinary	1.33 (0.87–1.82)	1.72 (1.14–2.39)	1.10 (0.42–1.62)	-	0.106	0.173
3-Aminoglutarate (×10^−2^)	Urinary	4.97 (3.03–8.78)	7.64 (5.35–10.38)	4.31 (2.84–8.18)	-	0.095	0.161
Hypotaurine (×10^−2^)	Urinary	0 (0–5.66)	0 (0–0.68)	1.94 (0–6.37)	↑	0.043	0.132
**Purine and pyrimidine metabolites**							
Adenosine (×10^−1^)	Urinary	0.91 (0.67–1.51)	1.31 (0.78–2.39)	0.90 (0.61–1.28)	-	0.108	0.171
Guanine (×10^−2^)	Urinary	6.55 (3.14–9.89)	7.87 (4.98–11.56)	6.05 (2.38–7.93)	-	0.129	0.181
Uracil (×10^−2^)	Urinary	0.17 (0–0.85)	0.59 (0–1.71)	0 (0–0.78)	-	0.232	0.291
**Saccharides**							
Mannose (×10^−1^)	Plasma	0.59 (0.32–0.89)	0.43 (0.20–0.69)	0.64 (0.34–0.91)	-	0.173	0.229
Psicose (×10^−2^)	Plasma	1.90 (1.27–4.20)	3.16 (1.79–5.00)	1.84 (0.93–3.37)	-	0.078	0.147
Sorbose (×10^−2^)	Plasma	1.14 (0–2.18)	2.14 (0.61–4.31)	0.98 (0–1.89)	↓	0.043	0.124
Sucrose (×10^−2^)	Plasma	0 (0–1.24)	1.69 (0–6.92)	0 (0–0.79)	↓	0.029	0.109
Sucrose	Urinary	0.18 (0.10–0.38)	1.08 (0.04–1.20)	0.18 (0.11–0.33)	-	0.250	0.306
Tagatose (×10^−1^)	Urinary	0.38 (0.18–1.19)	0.93 (0.10–1.37)	0.36 (0.18–0.89)	-	0.334	0.372
Xylose (×10^−1^)	Urinary	1.17 (0.55–1.79)	1.20 (0.36–1.65)	1.16 (0.59–1.94)	-	0.510	0.510
**Sugar alcohols**							
Mannitol	Plasma	1.04 (0.87–1.26)	0.94 (0.78–1.03)	1.10 (0.91–1.35)	↑	0.033	0.116
Sorbitol (×10^−1^)	Plasma	1.31 (0.94–1.97)	1.95 (0.81–2.40)	1.26 (0.95–1.90)	-	0.305	0.356
Galactitol (×10^−1^)	Urinary	1.49 (0.79–2.44)	1.64 (1.15–4.71)	1.40 (0.78–2.37)	-	0.214	0.276
**Urinary monocarboxylates**							
Lactate	Urinary	0.86 (0.64–1.15)	0.69 (0.55–0.91)	0.97 (0.68–1.17)	↑	0.045	0.123
Nicotinate (×10^−2^)	Urinary	3.42 (1.20–7.61)	0.99 (0.35–1.37)	5.42 (1.97–8.78)	↑	0.001	0.025
**Others**							
3-Hydroxypropionate (×10^−2^)	Plasma	7.08 (6.32–8.13)	6.45 (5.63–8.04)	7.38 (6.42–8.13)	-	0.334	0.364
Phosphorate	Plasma	20.8 (19.0–24.3)	23.9 (20.6–27.6)	20.2 (18.2–22.5)	↓	0.024	0.098
Pyrogallol (×10^−3^)	Plasma	0 (0–0)	0 (0–5.30)	0 (0–0)	↓	0.006	0.059
3-Methoxy-4-hydroxybenzoate (×10^−2^)	Urinary	1.06 (0–1.80)	0.77 (0–1.08)	1.29 (0–1.95)	-	0.094	0.165
Creatinine	Urinary	0.65 (0.41–1.03)	1.02 (0.53–2.00)	0.60 (0.40–0.91)	-	0.055	0.117
Hydroxylamine	Urinary	0.84 (0.65–1.00)	0.67 (0.50–0.79)	0.90 (0.70–1.05)	↑	0.007	0.057
Malonate (×10^−2^)	Urinary	1.63 (1.04–2.29)	1.27 (1.01–2.03)	1.68 (1.12–2.67)	-	0.263	0.314
Paraxanthine (×10^−2^)	Urinary	3.78 (1.52–8.06)	1.53 (0.33–3.38)	4.67 (2.68–8.45)	↑	0.016	0.078

Abbreviations: IQR, interquartile range; AHU, asymptomatic hyperuricemia. Raw *p*-values between the asymptomatic hyperuricemia and gout groups were calculated using the Mann–Whitney U test (without Benjamini–Hochberg adjustment). Adjusted *p*-values were calculated using the Benjamini–Hochberg method.

**Table 5 biomedicines-12-00300-t005:** Potential biomarkers distinguishing asymptomatic hyperuricemia from high-risk gout (SUA ≥ 9.0 mg/dL).

		Overall Gout		Low-Risk Gout (SUA < 9.0 mg/dL)		High-Risk Gout (SUA ≥ 9.0 mg/dL)	
	Sample	Fold Difference	Adjusted *p*	Fold Difference	Adjusted *p*	Fold Difference	Adjusted *p*
**Glucose metabolites**							
Glucose	Plasma	1.264	**0.074**	1.346	0.387	1.199	**0.129**
Pyruvate	Plasma	1.167	**0.124**	1.113	0.596	1.327	**0.116**
Lactate	Plasma	1.252	**0.080**	1.250	0.389	1.291	**0.137**
1,5-Anhydroglucitol	Plasma	0.628	**0.121**	0.698	0.470	0.616	0.170
2-Phosphoglycerate	Urinary	–	0.180	–	0.445	–	0.526
**TCA cycle intermediates**							
Aconitate	Plasma	–	**0.127**	–	0.497	–	**0.128**
Isocitrate	Plasma	1.238	**0.092**	1.153	0.477	1.369	**0.111**
2-Ketoglutarate	Plasma	1.415	**0.039**	1.277	0.336	1.566	**0.016**
Malate	Plasma	1.193	0.372	1.198	0.910	1.398	**0.149**
**Lipid metabolites**							
Arachidonate	Plasma	0.788	**0.082**	0.897	0.443	0.656	**0.065**
Eicosapentaenoate	Plasma	0.771	**0.091**	0.807	0.428	0.612	**0.065**
Docosahexaenoate	Plasma	2.273	0.179	2.818	0.469	1.927	0.552
Caproate	Plasma	0.900	0.365	0.985	0.962	0.838	0.164
Nonanoate	Plasma	0.484	**0.065**	0.675	0.394	0.432	**0.031**
**Amino acid metabolites**							
Tryptophan	Plasma	0.703	**0.128**	0.797	0.443	0.696	0.185
5-Oxoproline	Plasma	0.883	**0.131**	0.883	0.411	0.883	0.230
Serine	Plasma	0.429	0.151	0.434	0.408	0.421	0.327
Cysteine	Plasma	1.248	**0.123**	0.223	0.450	0.280	0.539
Lysine	Plasma	0.932	0.364	0.898	0.478	1.085	0.975
Tyramine	Plasma	0.906	0.374	0.866	0.500	1.016	0.983
Aspartate	Urinary	0.609	0.181	0.945	0.863	0.483	**0.129**
Tyramine	Urinary	0.626	0.197	0.791	0.796	0.495	0.180
4-Aminobutyrate	Urinary	0.494	**0.125**	0.581	0.485	0.245	**0.130**
5-Aminolevulinate	Urinary	0.640	0.173	0.804	0.798	0.5	**0.143**
3-Aminoglutarate	Urinary	0.564	0.161	0.707	0.815	0.436	**0.135**
Hypotaurine	Urinary	–	**0.132**	–	0.463	–	0.169
**Purine and pyrimidine metabolites**							
Adenosine	Urinary	0.687	0.171	0.763	0.715	0.542	0.172
Guanine	Urinary	0.769	0.181	0.848	0.789	0.555	0.171
Uracil	Urinary	–	0.291	0.983	0.891	–	0.243
**Saccharides**							
Mannose	Plasma	1.488	0.229	1.430	0.479	1.280	0.490
Psicose	Plasma	0.582	**0.147**	0.661	0.526	0.544	0.178
Sorbose	Plasma	0.458	**0.124**	0.621	0.528	0.244	**0.123**
Sucrose	Plasma	-	**0.109**	-	0.359	-	0.288
Sucrose	Urinary	0.167	0.306	0.944	0.583	1.056	0.593
Tagatose	Urinary	0.387	0.372	0.398	0.801	0.290	0.534
Xylose	Urinary	0.967	0.510	1.208	0.462	0.667	0.988
**Sugar alcohols**							
Mannitol	Plasma	1.170	**0.116**	1.213	0.353	1.074	0.336
Sorbitol	Plasma	0.646	0.356	0.913	0.962	0.579	0.281
Galactitol	Urinary	0.854	0.276	0.872	0.686	0.689	0.422
**Urinary monocarboxylates**							
Lactate	Urinary	1.406	**0.123**	1.333	0.435	1.449	0.173
Nicotinate	Urinary	5.475	**0.025**	3.616	0.380	6.515	**0.020**
**Others**							
3-Hydroxypropionate	Plasma	1.144	0.364	1.220	0.426	1.051	0.999
Phosphorate	Plasma	0.845	**0.098**	0.851	0.432	0.831	**0.146**
Pyrogallol	Plasma	–	**0.059**	–	0.456	–	0.186
3-Methoxy-4-hydroxybenzoate	Urinary	1.675	0.165	2.039	0.343	1.221	0.741
Creatinine	Urinary	0.588	**0.117**	0.627	0.429	0.559	0.211
Hydroxylamine	Urinary	1.343	**0.057**	1.322	0.412	1.371	**0.067**
Malonate	Urinary	1.323	0.314	1.724	0.380	1.024	0.984
Paraxanthine	Urinary	3.052	**0.078**	3.052	0.882	2.961	0.237

Abbreviations: SUA, serum urate levels. Fold differences were determined by comparing medians between the groups. *p*-values were calculated using Steel’s test with adjustment using the Benjamini–Hochberg method to control the false discovery rate. Fold differences with a median of 0 (below the detection limit by GC-MS/MS) in any of the groups are denoted by “–”.

## Data Availability

The datasets generated or analyzed during the current study are available from the corresponding author upon reasonable request.

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
