# Peer review of "Plasma and Urinary Metabolomic Analysis of Gout and Asymptomatic Hyperuricemia and Profiling of Potential Biomarkers: A Pilot Study"

_biomedicines, 2024, doi:10.3390/biomedicines12020300_

Round 1

Reviewer 1 Report

Comments and Suggestions for Authors

Dear Editor,

I carefully read the manuscript "Plasma and Urinary Metabolomic Analysis of Gout and Asymptomatic Hyperuricemia and Profiling of Potential Biomarkers: A Pilot Study", that reports data from an outstanding proof-of-concept study including 33 patients with gout and 9 individuals with asymptomatic hyperuricemia.

My (few) comments and suggestions for the authors are the following:

 - The authors should highly consider to refer to doi: 10.3390/medicina57010058 and findings from URRAH Study (doi: 10.1007/s40292-023-00602-4) in their manuscript.

 - Line 91: Why did the authors choose to include only male individuals in their analysis?

 - English language needs to be carefully revised and improved. For example, "Basic" (line 94) does not mean anything.

 - The limitations of the study should be further and more deeply discussed by the authors. Among the limitations, the authors should mention on the small sample size.

Comments on the Quality of English Language

English language needs to be carefully revised and improved.

Reviewer 2 Report

Comments and Suggestions for Authors

A well written manuscript describing preliminary metabolomic And genetic data on hyperuricemic individuals some of whom have gout and high uric acid values compared to a group also with gout but lower uric acid, and Asymptomatic hyperuricemia group. Not sure why a uric acid value of 9 mg percent was chosen given that the considered upper limit of normal of uric acid is 7 mg percent in men and 6 mg percent in women. While I am not sure of the IMPACT, other comorbidities such as hypertension and diabetes as well as treatment for these medical problems as well as Urate lowering medication should be described in the baseline characteristics of the population. it would probably be useful to extend a training model to a validation model. To the authors credit they have described many of the limitations of their preliminary data. I should note that most of this data is not particularly new and has been described in many of the references in terms of aberrant metabolic pathways in the development of hyperuricemia and gout. Curious why the hematocrit was lower in the hyperuricemic doubt population. Could this have been a reflection of renal impairment?

Comments on the Quality of English Language

Excellent quality language.

Reviewer 3 Report

Comments and Suggestions for Authors

The article is devoted to the study of the current problem of uric acid metabolism disorders and the development of gout.

The authors studied this problem using metabolomics analysis. This is one of the new, modern and labor-intensive approaches to the study of fundamental molecular biochemical factors and mechanisms of disease development.

The research carried out is very interesting, well and efficiently conducted at a high level using modern high-tech research methods. An in-depth analysis of the results obtained in the “Discussion” section indicates that the authors are well versed in the problem of gout.

I have several comments that will help improve the article.

1. The authors showed that subgroups of men differed by age (Table 2). Was the effect of patient age on the results obtained taken into account? Were the results standardized by age factor? This needs to be specified.

2. Patients were included in the study between September 2019 and April 2020. At this time, there was already a Covid 19 pandemic. Was the fact of illness in patients taken into account? This could have influenced the results of the study. This needs to be specified.

3. The number of patients is not large. You need to make a section called “Limitations of the Study” and include some factors in it.

Overall, I have a positive review. The article will be well read and cited by many scientists studying the problem of uric acid metabolism disorders and the development of gout.

Round 2

Reviewer 2 Report

Comments and Suggestions for Authors

Thank you for your responses to the comments. Again, the only significant criticism is the absence of a validation set which would add appreciably to the validity of the findings.